# SumRA: Parameter Efficient Fine-tuning with Singular Value Decomposition and Summed Orthogonal Basis

**Chin Yuen Kwok**[1,2], **Yongsen Zheng**[1,2]*, **Jia Qi Yip**[2], **Kwok-Yan Lam**[1,2], **Eng Siong Chng**[2]

[1]Digital Trust Centre, Nanyang Technological University, Singapore
[2]College of Computing and Data Science, Nanyang Technological University, Singapore
{kwok0062, jiaqi006}@e.ntu.edu.sg
{yongsen.zheng, kwokyan.lam, aseschng}@ntu.edu.sg

## Abstract

Parameter-efficient fine-tuning (PEFT) aims to adapt large pretrained speech models using fewer trainable parameters while maintaining performance. Low-Rank Adaptation (LoRA) achieves this by expressing the weight update to a pretrained matrix $W_0$ as the product of two low-rank matrices, $A$ and $B$, yielding the adapted weight $W' = W_0 + BA$. Previous studies showed that freezing $A$ and only updating $B$ can reduce trainable parameters and achieve performance close to standard LoRA, where $A$ is initialized using the principal singular vectors of $W_0$ obtained via singular value decomposition (SVD). However, because $A$ is typically initialized with only the leading singular vectors, its representation capacity is confined to a narrow subspace of the model's knowledge. To overcome this limitation, we propose SumRA, which initializes each row of $A$ as a sum of multiple singular vectors chosen from beyond the leading components, thereby enabling $A$ to influence a larger portion of the model's knowledge space. Experiments on multilingual automatic speech recognition (ASR) tasks show that by adapting Whisper to five new languages from Common Voice with only 10 hours of data each, our method improves word error rate from 14.42% to 12.41% over LoRA baselines while using 50% less trainable parameters.

## 1 Introduction

Parameter-Efficient Fine-Tuning (PEFT) provides an effective strategy for adapting large pretrained models to diverse downstream tasks while minimizing additional parameters and computational overhead Wang et al. (2024); Chu et al. (2023); Hu et al. (2024b). Among PEFT methods, Low-Rank Adaptation (LoRA) (Hu et al., 2022; 2024a) is the most classical approach, adapting large speech models by inserting and updating only small low-rank matrices (Fig. 1B). This design substantially reduces the storage cost of additional parameters (Lin et al., 2024)—for example, a single LoRA for a 1B-parameter model typically requires only tens of megabytes. However, when scaling to thousands (Brüel-Gabrielsson et al., 2024) or even millions (Parada et al., 2025) of LoRAs, total storage can still reach up to 10 TB, posing significant scalability challenges. This issue is especially critical in multilingual (Della Libera et al., 2024) and personalized (Joseph & Baby, 2024) Automatic Speech Recognition (ASR), where each user requires a dedicated LoRA tailored to their language proficiency, accent, and domain-specific vocabulary (Kwok et al., 2025; Shim et al., 2024; Rolland & Abad, 2024). At scale, a large user base can therefore require thousands or millions of LoRAs, greatly amplifying storage and deployment costs.

Multiple lines of work have explored improvements to LoRA. AdaLoRA Zhang et al. (2023b) dynamically adjusts (and progressively reduces) the rank of the LoRA matrices during training. LoRA+ Hayou et al. (2024) improves LoRA by applying differential learning rates to the projection matrices $A$ and $B$. Complementary to these methods, LoRA-FA (Zhang et al., 2023a) reduces the

---

*Corresponding author.

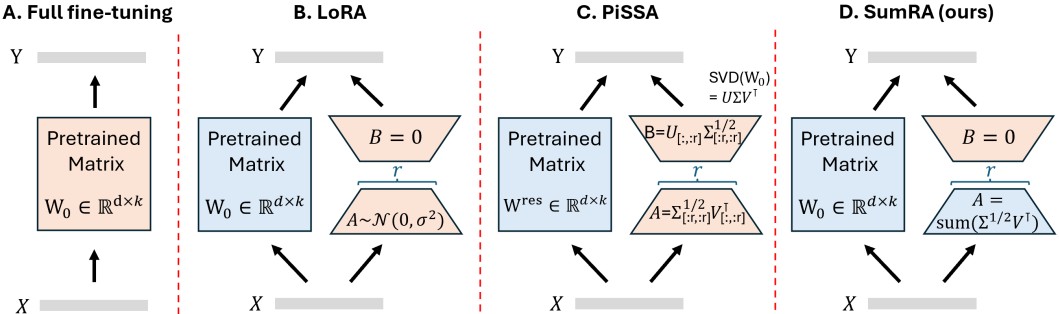

Figure 1: Comparison of full fine-tuning, LoRA, PiSSA, and our SumRA method. Only the pre-trained matrix $W_0$ of a linear layer is shown for simplicity. Blue modules indicate frozen parameters, orange modules are trainable. A) Full fine-tuning updates all of $W_0$; B) LoRA freezes $W_0$ and trains two low-rank matrices; C) PiSSA initializes $A$ and $B$ from singular vectors; D) SumRA enhances PiSSA with a novel initialization of $A$ (see Fig. 2 and 3).

trainable parameters by freezing the projection-down matrix $A$ and updating only the projection-up matrix $B$, thus improving efficiency and reducing memory usage. Nevertheless, LoRA-FA typically initializes $A$ randomly, missing the opportunity to leverage the pretrained knowledge embedded in $W_0$. To address this, PiSSA (Meng et al., 2024) proposes a more principled initialization approach: unlike LoRA which initializes $A$ from normal distributions (Fig. 2A), PiSSA initializes $A$ using the leading singular vectors from the singular value decomposition (SVD) of $W_0$ (Fig. 2B). This allows the low rank matrices to be fine-tuned in the direction that $W_0$ has the most significant stretching or impact.

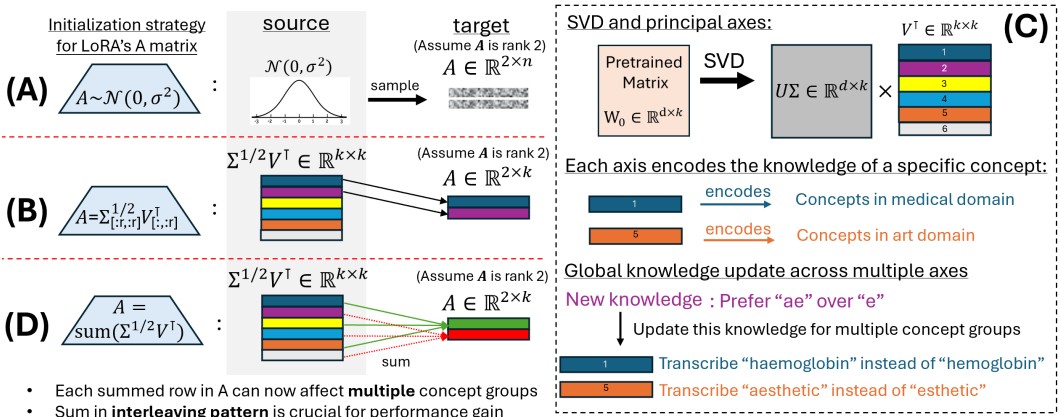

Figure 2: Initialization strategies for LoRA's $A$ matrix. Pre-trained weight matrix is $W_0 \in \mathbb{R}^{m \times n}$ and $r$ is LoRA's rank. $\text{SVD}(W_0) = U\Sigma V^T$. A) The $A$ matrix is initialized from normal distribution. B) The $A$ matrix is initialized from the top $r$ rows (singular vectors) in $\Sigma^{1/2}V^T$. C) Assume that each singular vector encodes knowledge relevant to a specific subset of the speech model's vocabulary. Assigning a single singular vector to each row of A limits that row's influence to one subset at a time. D) To address this, we propose to sum multiple rows of $\Sigma^{1/2}V^T$ into each row of $A$.

In PiSSA initialization, each singular vector extracted from $W_0$ encodes only knowledge associated with a specific concept (Merullo et al., 2024; Pan et al., 2024). Since PiSSA initializes $A$ using only the top few singular vectors—typically less than 5% of all available ones (Yang et al., 2024)—the resulting initialization restricts $A$'s influence to a narrow set of concept groups. Consequently, the adaptation primarily targets a limited portion of the model's learned knowledge. To overcome this limitation, we propose summing multiple singular vectors into each row of $A$, as illustrated in Fig. 2D. By aggregating vectors, $A$ gains the capacity to represent a broader range of concepts, thereby extending its influence to a larger portion of the model's knowledge space. This

enhances the effectiveness of adaptation, particularly for tasks that require global rather than localized knowledge transfer. By summing singular vectors, computations that are originally performed on the individual vectors are combined. This mechanism of sharing model computations across multiple concepts or dimensions has demonstrated effectiveness in related work (Reneau et al., 2025). To improve the effectiveness of the vector aggregation, we further propose two summing strategies—interleave sum and greedy sum (Figs.3B and C)—that distribute singular vectors across rows so that the sum of singular values in each row is balanced. This ensures that highly important vectors are evenly assigned rather than concentrated in a single row, reducing interference between summed vectors. A comparison between LoRA and our SumRA is provided in Table1.

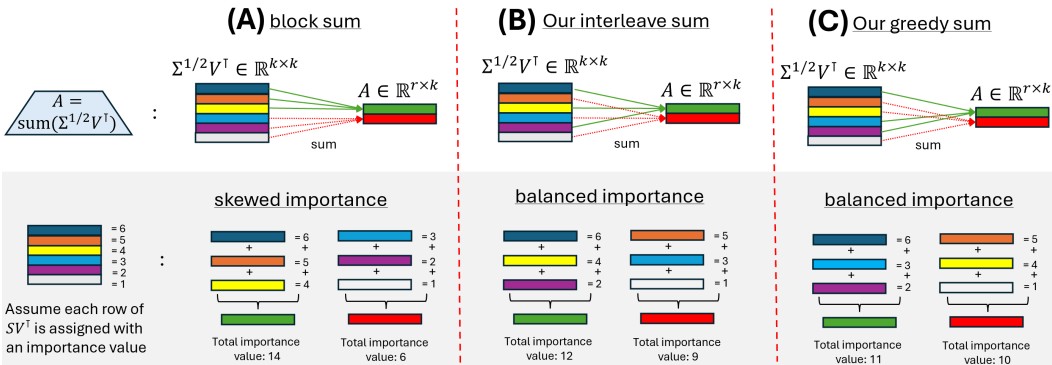

Figure 3: Strategies to sum $\Sigma^{1/2}V^T$ into matrix $A$. To improve global knowledge update as discussed in Fig. 2, we propose to compress and sum all singular vectors into $A$. However, summing the most important singular vectors to the same row as shown in A) block sum is unideal as it causes destructive interference between them. Therefore, we propose B) interleave sum and C) greedy sum that evenly assign the important singular vectors to the rows of $A$.

Table 1: Comparison between LoRA and our SumRA method to update a linear layer's pretrained weight $W_0 \in \mathbb{R}^{d \times k}$. $h$ and $x$ are the input and outputs of the linear layer respectively. $r$ is the rank of the low rank matrices and $\Delta W$ is the weight update. In this table, **bold** highlights the model's trainable component during adaptation. SVD is performed on $W_0$ to obtain $U$, $\Sigma$ and $V$.

|  | LoRA | our SumRA |
|---|---|---|
| Forward | $h = (W_0 + \Delta W)x = (W_0 + \mathbf{B}A)x$ | $h = (W_0 + \Delta W)x = (W_0 + \mathbf{B}A)x$ |
| Initialization | $\mathbf{B} = 0 \in \mathbb{R}^{d \times r}$ 

 $\mathbf{A} \sim \mathcal{N}(0, \sigma^2) \in \mathbb{R}^{r \times k}$ | $\mathbf{B} = 0 \in \mathbb{R}^{d \times r}$ 

 $H = \mathrm{reshape}(\Sigma^{1/2}V^\top, \mathbb{R}^{k \times k} \to \mathbb{R}^{\frac{k}{r} \times r \times k})$ 

 $A = \mathrm{sum}(H, \dim = 0) \in \mathbb{R}^{r \times k}$ |
| Comparison | Fine-tune both $\mathbf{B}$ and $\mathbf{A}$ 
 Store both $\mathbf{B}$ and $\mathbf{A}$ | Fine-tune $\mathbf{B}$ while freezing $A$ 
 Store separate $\mathbf{B}$s 
 Share $A$s for each task |

Empirical results on low-resource multilingual ASR tasks validate the effectiveness of SumRA. By adapting Whisper to five new languages from Common Voice—each with only 10 hours of data—SumRA achieves a 50% reduction in trainable parameters by freezing $A$, yet consistently outperforms LoRA, LoRA-FA and PiSSA, achieving up to an 16% relative improvement in word error rate despite using 50% less memory to store additional parameters. These findings highlight the value of structured, knowledge-aware initialization strategies for scalable and effective PEFT in multilingual and low-resource settings as shown in Fig. 4.

## 2 PRELIMINARY

This section reviews PEFT techniques relevant to our work. We begin with LoRA (Section 2.1) and its memory-optimized variant LoRA-FA (Secion 2.2), both designed to reduce the number of train-

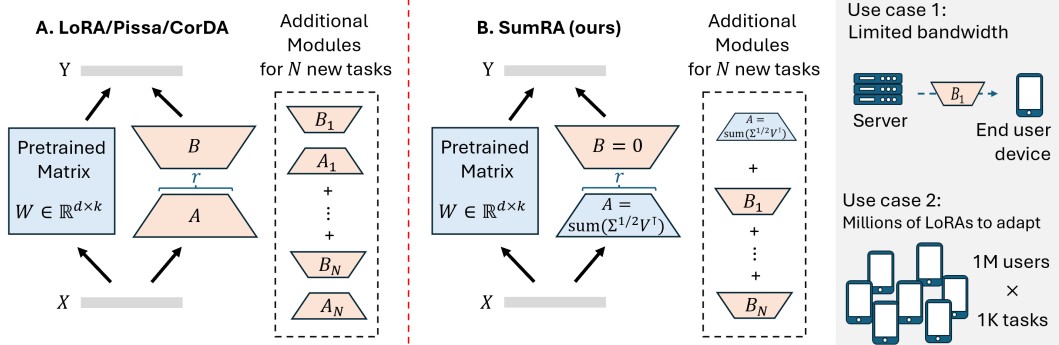

Figure 4: Memory cost comparison of LoRA/PiSSA/Corda with our SumRA method. Blue/orange modules are frozen/updated. Storage cost is reduced as instead of A) having a separate trainable $A$ for each task, B) the same frozen $A$ can be shared between different tasks in SumRA.

able parameters. We then introduce PiSSA (Secion 2.3) and CorDA (Secion 2.4), which improve LoRA's initialization using SVD to enhance adaptation efficiency and task alignment.

## 2.1 LoRA

Inspired by findings on the low intrinsic dimensionality of language models (Aghajanyan et al., 2020), LoRA constrains weight updates to a low-rank subspace to reduce the number of trainable parameters during fine-tuning. Given a pre-trained weight matrix $W_0 \in \mathbb{R}^{d \times k}$, LoRA approximates the update as $W_0 + \Delta W = W_0 + BA$, where $B \in \mathbb{R}^{d \times r}$ and $A \in \mathbb{R}^{r \times k}$ with $r \ll \min(d, k)$. During training, $W_0$ remains frozen and only $A$, $B$ are updated. The forward computation becomes:

$$h = W_0 x + \frac{\alpha}{r} BA x, \tag{1}$$

where $h \in \mathbb{R}^d$ and $x \in \mathbb{R}^k$ are the input and output activations, and $\alpha$ is a constant in $r$. Tuning $\alpha$ is roughly the same as tuning the learning rate of LoRA (Hu et al., 2022).

## 2.2 LoRA-FA

While LoRA reduces the number of trainable parameters, it still incurs substantial memory overhead in fine-grained, personalized settings (Parada et al., 2025). LoRA-FA (Zhang et al., 2023a) further reduces memory by freezing both $W_0$ and $A$ in LoRA, updating only $B$. In this setup, $A$ is randomly initialized from a normal distribution and remains fixed, while $B$ is initialized to zero, similar to standard LoRA.

This design is inspired by findings that LoRA can be trained effectively using fixed, random basis (Koohpayegani et al., 2023). Given than A is frozen, and assuming an RQ decomposition $A = RQ$ and defining $\bar{B} = BR$, the low-rank term $BAx$ in Eq. 1 becomes:

$$BAx = \bar{B}Qx, \tag{2}$$

where $Q$ is an orthonormal matrix. In this formulation, training $B$ effectively tunes $\bar{B}$ to operate on features projected by the fixed, random basis in $Q$, enabling efficient adaptation with minimal memory usage.

## 2.3 PiSSA

Building on LoRA, PiSSA improves the initialization of the adapter matrices $A$ and $B$ using the principal singular vectors of the pre-trained weight matrix $W_0$. Given the SVD of $W_0 \in \mathbb{R}^{d \times k}$ as $W_0 = USV^\top$, PiSSA uses the top-$r$ components to initialize:

$$B = U_{[:,:r]}\Sigma_{[:r,:r]}^{1/2} \in \mathbb{R}^{d \times r} \tag{3}$$

$$A = \Sigma_{[:r,:r]}^{1/2}V_{[:,:r]}^\top \in \mathbb{R}^{r \times k} \tag{4}$$

The remaining components form the residual matrix:
$$W^{res} = U_{[:,r:]}\Sigma_{[r:,r:]}V_{[:,r:]}^{\top}, \tag{5}$$
which is frozen during fine-tuning. This preserves the original model output at initialization:
$$h = W_0 x = (W^{res} + BA)x, \tag{6}$$
Compared to LoRA, PiSSA updates the principal components while freezing the "residual" parts, allowing faster convergence and enhanced performance.

## 2.4 CORDA

While PiSSA performs SVD on the pre-trained weight $W_0$, CorDA rescales $W_0$ before applying SVD. The core idea is to compute the covariance matrix $C = XX^{\top}$ from input activations $X \in \mathbb{R}^{k \times n}$ collected from $n$ target task data samples, and preform SVD on $W_0 C$, which is a rescaled version of $W_0$ oriented by the target task data distribution:
$$\text{SVD}(W_0 C) = U\Sigma V^{\top} \tag{7}$$
To preserve initial model behavior, the weight $W_0$ is reconstructed as $\hat{W}$:
$$\hat{W} = \text{SVD}(W_0 C)C^{-1} = U\Sigma V^{\top}C^{-1} = U\Sigma\hat{V}^{\top} \tag{8}$$
with $\hat{V}^{\top} = V^{\top}C^{-1}$. If $C$ is not invertible, $\epsilon$ is added to its diagonal. This decomposition allows the extracted principal singular vectors to be aligned with the task-relevant directions in the activation space and allows for more targeted adaptation.

## 3 METHOD

Building upon PiSSA and CorDA, we first present our SumRA method as an improved LoRA matrix initialization strategy. It initializes each row of $A$ as a sum of multiple singular vectors obtained from the SVD of $W_0$ (Section 3.1). Following that, three strategies are presented to distribute singular vectors across rows of A to improve performance in Section 3.2.

### 3.1 SUMMING MULTIPLE SINGULAR VECTORS

The intuition behind our method is as follows. PiSSA demonstrates that initializing the low-rank matrices in LoRA using the top singular vectors of a pretrained weight matrix $W_0$ guides fine-tuning along directions where $W_0$ has the most significant impact—i.e., directions associated with the largest singular values—resulting in improved performance. However, this approach may still be suboptimal, as (Staats et al., 2024) argue that singular vectors corresponding to smaller singular values can also be important for adaptation. Ideally, the $A$ matrix in LoRA should be initialized using all singular vectors to reduce information loss. However, it is infeasible to assign each singular vector to a unique row of $A$ as the rank of $A$ is usually smaller than the rank of $W_0$. To address this, we propose a strategy that sums multiple singular vectors into each row of $A$. This allows us to compress the full set of singular vectors into a lower rank matrix without discarding potentially useful directions for adaptation.

Specifically, our method first performs SVD on the pre-trained weight $W_0$ similar to PiSSA:
$$\text{SVD}(W_0) = U\Sigma V^{\top} = \sum_{i=1}^{k} \sigma_i \mathbf{u}_i \mathbf{v}_i^{\top}. \tag{9}$$
where $\sigma_i$ is the $i$-th diagonal element in $\Sigma$, $\mathbf{u}_i$ is the $i$-th column in $U$ and $\mathbf{v}_i$ is the $i$-th column of $V$. Then, given $\mathbf{a}_i \in \mathbb{R}^k$ is the $i$-th row of $A \in \mathbb{R}^{r \times k}$, and $S_i$ be the subset of column indexes of $V$ that is assigned to $\mathbf{a}_i$, we initialize $\mathbf{a}_i$ as:
$$\mathbf{a}_i = \sum_{j \in S_i} \sqrt{\sigma_j}\mathbf{v}_j. \tag{10}$$

To compress the full set of singular vectors into $A$, each column index of $V$ must be assigned to exactly one of the subset $S_i$ such that $\bigcup_{1 \leq i \leq r} S_i = \{1, \ldots, k\}$. Furthermore, all the subsets should be disjoint such that $S_i \cap S_j = \emptyset$ for all $1 \leq i, j \leq r$ and $i \neq j$. This allows all rows of $A$ to be orthogonal to each other such that $\mathbf{a}_i \perp \mathbf{a}_j$ for all $1 \leq i, j \leq r$ and $i \neq j$. The orthogonality constraint improves the optimization efficiency during model fine-tuning Cao & Song (2024).

## 3.2 SUMMATION STRATEGY

However, it is inevitable that there will be interference (information loss) between the singular vectors if they are summed together. To prevent the destructive interference between the more important vectors (estimated by the singular values), they should not be summed into the same row as shown in Fig. 3A.

To address this, we compare three summing strategies. The first strategy is naive block sum as shown in Fig. 3A. Specifically, assume $k$ is divisible by $r$, this strategy assigns singular vectors by having:

$$S_i = \{ \frac{(i-1)k}{r} + 1, \frac{(i-1)k}{r} + 2, ..., \frac{ik}{r} \} \tag{11}$$

However, block sum is unideal as it assigns all the important singular vectors to the same row of $A$ and causes destructive interference between them. Instead, they should be evenly assigned to prevent interference. Specifically, define the load $L_j$ of $\mathbf{a}_j$ as the sum of the singular values of all singular vectors assigned to that row:

$$L_j = \sum_{i \in S_j} \sigma_i \tag{12}$$

The distribution is most even when the maximum row load $L_{\max} = \max_j L_j$ is minimized. To reduce $L_{\max}$, we propose interleave sum and greedy sum as shown in Fig. 3B and C that evenly assign the important singular vectors to the rows of $A$. For interleave sum, we have:

$$S_i = \{ 0\frac{k}{r} + i, 1\frac{k}{r} + i, \dots, (r-1)\frac{k}{r} + i \} \tag{13}$$

Compared to interleave sum, we further derived greedy sum, an optimal summation strategy that always results in the most even distribution. Specifically, Let there be $K$ singular vectors (scaled by $\Sigma$) in $\Sigma V^T$, each associated with a singular value, sorted in non-increasing order. Assume we are assigning the singular vectors to the rows of $A$ one by one. Then, our greedy sum strategy will at each step, assign the next singular vector (in decreasing order of singular value) to the row with the currently minimum load. We proof that greedy sum produces an assignment that minimizes $L_{\max}$ in Appendix A.1.

## 4 CONNECTION TO MODEL AVERAGING

Our approach, which sums multiple singular vectors together, is closely related to model averaging. Instead of compressing all singular vectors into a single low-rank matrix $A$, a naive alternative is to initialize multiple versions of $A$ using different subsets of singular vectors from $\Sigma^{1/2}V^\top$, and independently adapt the pre-trained model into multiple variants. Since each model is fine-tuned in a different direction—owing to distinct singular vector initialization—they may capture complementary aspects of the adapted task. Model averaging can then be applied to merge these models, leveraging their diverse adaptations for improved performance. As illustrated in Fig. 5, our SumRA interleave-sum method achieves a similar effect, but in a more efficient manner: we sum the LoRA matrices before adaptation, effectively integrating diverse adaptation directions within a single model.

## 5 EXPERIMENT

In this section, we begin by introducing the datasets and training configuration used for multilingual ASR adaptation in Sections 5.1, 5.2 respectively.We then perform a comprehensive evaluation of our SumRA method, where Section5.3 presents the main results on multilingual ASR adaptation and 5.4 analyze the performance of our method under different summation strategies as discussed in Section 3.2.

Our SumRA interleave sum method is closely related to model averaging:

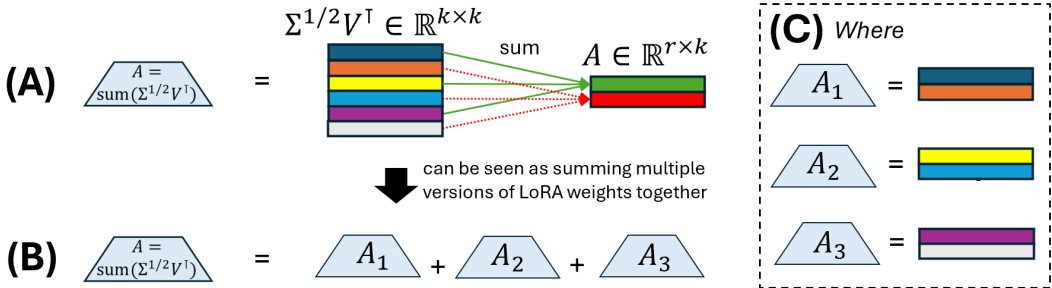

Figure 5: Comparison between our SumRA interleave sum method with model averaging. A) As an example, consider our SumRA method which interleave sum the matrix $\Sigma^{1/2}V^{\top} \in \mathbb{R}^{k \times k}$ to initialize the LoRA $A$ matrix. B) This can be seen as performing weight summation on three different LoRA $A$ matrices, where C) each of the matrix is initialized from a different part of $\Sigma^{1/2}V^{\top}$.

## 5.1 DATASET

Building on previous studies (Yu et al., 2023; Della Libera et al., 2024), we evaluate our approach using a subset of the widely-used Common Voice MASR dataset (Ardila et al., 2020). This publicly available dataset, collected via crowd-sourcing, contains short audio recordings paired with transcriptions across 108 languages, amounting to a total of 17,690 validated hours. The data is organized into training, validation, and test splits. Following the methodology of (Della Libera et al., 2024), we select five languages for our experiments, including Esperanto (eo), Meadow Mari (mhr), Central Kurmanji Kurdish (kmr), Frisian (fy-NL), and Interlingua (ia). These languages are not seen in the training set of our pre-trained model. For each language, we follow (Della Libera et al., 2024) to partition the training, validation, and test sets consisting of ten hours, one hour, and one hour of audio, respectively.

## 5.2 TRAINING CONFIGURATION

Whisper (Radford et al., 2023) is chosen as the MSAR model in our study. Whisper employs an encoder-decoder transformer architecture (Gulati et al., 2020), where the encoder processes input Mel spectrograms to extract audio features. The auto-regressive decoder then generates transcriptions conditioned on the encoder's hidden representations, following a multitask framework that leverages special tokens as task specifiers.

We adapt both small and large-v2 variants of the Whisper model to accommodate new languages. Model adaptation is performed over two epochs with a training batch size of 4. Validation is carried out every $1/8$ epoch across all methods. We use the AdamW optimizer (Loshchilov & Hutter, 2019) in conjunction with a variant of the ReduceLROnPlateau learning rate scheduler . LoRA is added to the linear layers of all the feed-forward and attention layers in the decoder. $\alpha$ for LoRA is equal to the rank following Lee et al. (Lee et al., 2023). Only the parameters of the LoRA-based modules and the normalization layers (Ba et al., 2016) are updated during adaptation. All experiments employ a greedy decoding strategy.

We compare our method against several LoRA-based adaptation baselines, including vanilla LoRA (Hu et al., 2022), VeRA (Kopiczko et al., 2023), LoRA-FA (Zhang et al., 2023a), DoRA (Liu et al., 2024), PiSSA (Meng et al., 2024), and CorDA (Yang et al., 2024). To further validate the effectiveness of these adaptation strategies, we also report results from the unadapted Whisper model and a fully fine-tuned (FT) Whisper model, where all parameters are updated during adaptation.

---

https://commonvoice.mozilla.org/en
https://speechbrain.readthedocs.io/en/latest/_modules/speechbrain/nnet/schedulers.html#NewBobScheduler

We evaluate our methods using word error rate (WER), which is a common metric used to evaluate the accuracy of ASR systems. It measures the percentage of words that are incorrectly recognized in a transcription compared to a reference (ground-truth) transcription.

Table 2: Comparison of SumRA with existing LoRA-based methods on the multilingual ASR adaptation task using WER. Lower is better. The best results are highlighted in bold, while the second-best results are underlined. The additional storage cost (param.) does not include weights that can be shared between tasks. The differences in WERs are shown in subscripts, relative to LoRA's WER.

| Method | Model | param. | rank | eo | ia | fy-NL | mhr | kmr |
|---|---|---|---|---|---|---|---|---|
| unadapted | | - | - | 94.37 | 74.53 | 1.05e+2 | 1.15e+2 | 1.14e+2 |
| FT | | - | - | 18.89 | 13.79 | 28.69 | 32.37 | 39.63 |
| LoRA | whisper-small | 0.5M | 2 | $28.76_{+0.0\%}$ | $19.99_{+0.0\%}$ | $50.81_{+0.0\%}$ | $53.75_{+0.0\%}$ | $60.10_{+0.0\%}$ |
| VeRA | | 0.2M | 2 | $38.99_{+66\%}$ | $27.42_{+79\%}$ | $67.81_{+72\%}$ | $68.08_{+68\%}$ | $76.42_{+58\%}$ |
| VeRA | | 0.3M | 768 | $30.54_{+6.2\%}$ | $21.30_{+6.6\%}$ | $55.63_{+9.5\%}$ | $56.60_{+5.3\%}$ | $63.71_{+6.0\%}$ |
| LoRA-FA | | 0.4M | 2 | $39.92_{+39\%}$ | $29.14_{+46\%}$ | $68.54_{+35\%}$ | $69.85_{+30\%}$ | $77.63_{+29\%}$ |
| DoRA | | 0.7M | 2 | $28.94_{+0.6\%}$ | $20.15_{+0.8\%}$ | $52.30_{+2.9\%}$ | $53.88_{+0.2\%}$ | $61.30_{+2.0\%}$ |
| PiSSA | | 0.5M | 2 | $29.20_{+1.5\%}$ | $19.23_{-3.8\%}$ | $49.72_{-2.2\%}$ | $51.29_{-4.6\%}$ | $\underline{58.68}_{-2.48}$ |
| CorDA | | 0.5M | 2 | $\underline{27.05}_{-6.0\%}$ | $\underline{18.96}_{-5.2\%}$ | $\underline{47.66}_{-6.2\%}$ | $\underline{51.08}_{-5.0\%}$ | $58.85_{-2.1\%}$ |
| SumRA (ours) | | 0.4M | 2 | $\mathbf{26.29}_{-8.6\%}$ | $\mathbf{17.23}_{-13\%}$ | $\mathbf{44.92}_{-12\%}$ | $\mathbf{48.49}_{-9.8\%}$ | $\mathbf{54.32}_{-9.6\%}$ |
| unadapted | | - | - | 94.37 | 74.53 | 1.05e+2 | 1.15e+2 | 1.14e+2 |
| FT | | - | - | 18.89 | 13.79 | 28.69 | 32.37 | 39.63 |
| LoRA | whisper-small | 7.7M | 32 | $23.39_{+0.0\%}$ | $15.31_{+0.0\%}$ | $39.34_{+0.0\%}$ | $40.63_{+0.0\%}$ | $48.51_{+0.0\%}$ |
| VeRA | | 0.2M | 32 | $36.93_{+28\%}$ | $25.66_{+28\%}$ | $65.52_{+29\%}$ | $63.98_{+19\%}$ | $73.74_{+23\%}$ |
| VeRA | | 0.3M | 768 | $30.54_{+6.2\%}$ | $21.30_{+6.6\%}$ | $55.63_{+9.5\%}$ | $56.60_{+5.3\%}$ | $63.71_{+6.0\%}$ |
| LoRA-FA | | 3.9M | 32 | $26.77_{+14\%}$ | $17.04_{+11\%}$ | $45.98_{+17\%}$ | $46.55_{+15\%}$ | $55.45_{+14\%}$ |
| DoRA | | 7.8M | 32 | $21.79_{-6.8\%}$ | $\mathbf{12.53}_{-18\%}$ | $37.07_{-5.8\%}$ | $38.95_{-4.1\%}$ | $\mathbf{43.18}_{-11\%}$ |
| PiSSA | | 7.7M | 32 | $21.79_{-6.8\%}$ | $13.95_{-8.9\%}$ | $34.93_{-11\%}$ | $38.24_{-5.9\%}$ | $46.23_{-4.7\%}$ |
| CorDA | | 7.7M | 32 | $\underline{21.12}_{-9.7\%}$ | $\underline{13.11}_{-14\%}$ | $\underline{34.35}_{-13\%}$ | $\underline{36.53}_{-10\%}$ | $44.85_{-7.5\%}$ |
| SumRA (ours) | | 3.9M | 32 | $\mathbf{20.77}_{-11\%}$ | $13.38_{-13\%}$ | $\mathbf{33.37}_{-15\%}$ | $\mathbf{36.30}_{-11\%}$ | $\underline{44.47}_{-8.3\%}$ |
| unadapted | | - | - | 62.54 | 47.96 | 1.05e+2 | 81.28 | 1.02e+2 |
| FT | | - | - | 15.59 | 13.20 | 26.05 | 30.60 | 36.86 |
| LoRA | whisper-large-v2 | 2.4M | 2 | $15.96_{+0.0\%}$ | $9.85_{+0.0\%}$ | $29.24_{+0.0\%}$ | $39.02_{+0.0\%}$ | $44.55_{+0.0\%}$ |
| VeRA | | 0.8M | 2 | $64.18_{+27\%}$ | $47.14_{+45\%}$ | $41.78_{+43\%}$ | $48.90_{+25\%}$ | $55.06_{+24\%}$ |
| VeRA | | 1.2M | 1280 | $17.18_{+7.6\%}$ | $11.15_{+13\%}$ | $32.88_{+12\%}$ | $39.15_{+0.3\%}$ | $46.29_{+4.1\%}$ |
| LoRA-FA | | 1.6M | 2 | $61.42_{+35\%}$ | $49.44_{+66\%}$ | $44.68_{+53\%}$ | $52.52_{+35\%}$ | $58.74_{+32\%}$ |
| DoRA | | 2.9M | 2 | $16.76_{+5.0\%}$ | $11.10_{+13\%}$ | $33.99_{+16\%}$ | $39.53_{+1.3\%}$ | $46.08_{+3.4\%}$ |
| PiSSA | | 2.4M | 2 | $\underline{15.12}_{-5.3\%}$ | $9.70_{-1.5\%}$ | $\underline{27.84}_{-4.8\%}$ | $36.40_{-6.7\%}$ | $\underline{38.93}_{-12.6\%}$ |
| CorDA | | 2.4M | 2 | $15.55_{-2.6\%}$ | $\mathbf{8.98}_{-8.8\%}$ | $27.94_{-4.5\%}$ | $\underline{34.83}_{-10\%}$ | $39.37_{-11.6\%}$ |
| SumRA (ours) | | 1.6M | 2 | $\mathbf{14.55}_{-8.8\%}$ | $\underline{9.30}_{-5.6\%}$ | $\mathbf{25.83}_{-12\%}$ | $\mathbf{34.72}_{-11\%}$ | $\mathbf{38.63}_{-13\%}$ |
| unadapted | | - | - | 62.54 | 47.96 | 1.05e+2 | 81.28 | 1.02e+2 |
| FT | | - | - | 15.59 | 13.20 | 26.05 | 30.60 | 36.86 |
| LoRA | whisper-large-v2 | 34.3M | 32 | $14.42_{+0.0\%}$ | $8.67_{+0.0\%}$ | $24.75_{+0.0\%}$ | $32.39_{+0.0\%}$ | $37.72_{+0.0\%}$ |
| VeRA | | 0.8M | 32 | $20.68_{+43\%}$ | $15.83_{+83\%}$ | $43.60_{+76\%}$ | $49.29_{+52\%}$ | $57.73_{+53\%}$ |
| VeRA | | 1.2M | 1280 | $17.18_{+7.6\%}$ | $11.15_{+13\%}$ | $32.88_{+12\%}$ | $39.15_{+0.3\%}$ | $46.29_{+4.1\%}$ |
| LoRA-FA | | 17.6M | 32 | $16.42_{+13\%}$ | $10.31_{+19\%}$ | $30.69_{+24\%}$ | $37.55_{+16\%}$ | $44.24_{+17\%}$ |
| DoRA | | 34.9M | 32 | $13.45_{-6.7\%}$ | $\underline{8.28}_{-4.5\%}$ | $23.38_{-5.5\%}$ | $29.67_{-8.4\%}$ | $35.59_{-5.7\%}$ |
| PiSSA | | 34.3M | 32 | $\underline{13.00}_{-9.9\%}$ | $8.82_{+1.7\%}$ | $\underline{22.43}_{-9.4\%}$ | $29.97_{-7.5\%}$ | $\underline{34.26}_{-9.2\%}$ |
| CorDA | | 34.3M | 32 | $13.13_{-9.0\%}$ | $9.18_{+5.9\%}$ | $22.96_{-7.2\%}$ | $\underline{29.20}_{-9.9\%}$ | $36.33_{-3.7\%}$ |
| SumRA (ours) | | 17.6M | 32 | $\mathbf{12.41}_{-14\%}$ | $\mathbf{8.17}_{-5.8\%}$ | $\mathbf{22.27}_{-10\%}$ | $\mathbf{27.19}_{-16\%}$ | $\mathbf{34.21}_{-9.3\%}$ |

## 5.3 Performance on Multilingual ASR

Table 2 shows the results of adapting whisper-small and whisper-large-v2 to five new languages using different LoRA-based methods and using different ranks (2 and 32). The results show that before adaptation, Whisper generally achieves close to 100% WER as it has never been pre-trained to transcribe the new languages. After adaptation, the WER drops significantly. For whisper-small, full-finetuning (FT) generally achieves better WER than the LoRA methods as all the model weights can be updated to accommodate new language knowledge. However, for larger models like whisper-

Table 3: Comparison of different summation strategies. WERs of adapting whisper-small using our SuMRA method with rank 32 is shown. Lower is better. The additional storage cost (param.) does not include weights that can be shared between tasks.

| Method | param. | eo | ia | fy-NL | mhr | kmr |
|---|---|---|---|---|---|---|
| LoRA | 7.7M | 23.39 | 15.31 | 39.34 | 40.63 | 48.51 |
| SumRA (block sum) | 3.9M | 21.68 | 13.91 | 35.38 | 37.35 | 47.30 |
| SumRA (interleave sum) | 3.9M | 20.77 | 13.38 | **33.37** | **36.30** | **44.47** |
| SumRA (greedy sum) | 3.9M | **20.73** | **13.16** | 33.91 | 37.53 | 44.72 |

large-v2, adaptation is more prone to overfitting, and some LoRA-based methods performs better than FT as they can reduce overfitting by constraining weight updates.

For LoRA-based methods using rank 2, Vera, LoRA-FA and DoRA generally perform worse than LoRA as they are more restrictive on the weight update than LoRA. As the rank is increased to 32, the trainable parameters increases and the adaptation is more prone to overfitting than in the rank 2 case. As a result, DoRA generally performs better than LoRA for rank 32 as it can better reduce overfitting. Next, PiSSA, CorDA and SumRA generally performs better than LoRA in all cases as we hypothesize that the methods can reduce overfitting by initializing the LoRA matrices with singular vectors, and do not reduce the learning capacity of the matrices as no additional weight update constraints are imposed during training.

Finally, the results show that our SumRA method generally outperforms all other baseline LoRA-based methods while have low memory cost. This is because our method can share the LoRA $A$ matrices between tasks, thus adding more tasks does not increase the memory cost of the $A$ matrices. Furthermore, our SumRA method can outperform LoRA by up to 14%, and outperform CorDA by up to 11%, showing the effectiveness of our initialization strategy for LoRA matrix $A$.

## 5.4 SUMMATION STRATEGY ANALYSIS

We further perform ablation study to study the effect of different summation strategies. The results in Table 3 shows that block sum performs worse than all other summation strategies as it sums all the important singular vectors into the same row of $A$. Our interleave sum and greedy sum methods show consistent improvements over block sum as they can distribute the important singular vectors more evenly across the rows of $A$, validating the effectiveness of our summation strategies.

## 5.5 SUMRA'S EFFECT UNDER DIFFERENT DATA SCALES

To broaden the experimental scope, we evaluate SumRA under different data regimes (10h, 50h, and 100h) when adapting whisper-small to Esperanto, a new language with a larger available training corpus. Table 4 reports the resulting WER. Compared to LoRA, SumRA reduces WER from 23.39% to 20.77% in the 10h setting and from 15.20% to 14.49% in the 50h setting. In contrast, the improvement diminishes at 100h. These results indicate that SumRA provides more substantial gains in low-resource scenarios. Furthermore, the results in Table 4 row four shows that additional WER improvement is achieved when the previously frozen down-projection matrix $A$ (as shown in Fig. 1D) is trained, indicating that further performance gains can be obtained by updating the parameters in $A$. However, this improvement comes at the cost of increased trainable parameters and higher training overhead.

## 6 LIMITATION

Our approach is most effective for adapting global attributes, such as accents or speaking styles, that influence a large portion of the vocabulary. In these scenarios, summing multiple singular vectors enables coordinated parameter updates across many vocabulary subsets, providing clear advantages over methods that adapt only a small fraction of vectors. However, for local adaptations—such as

Table 4: WER (%) on Esperanto under different training data scales. Lower is better.

| Method | 10h | 50h | 100h |
|---|---|---|---|
| FT | 18.89 | 15.31 | 13.62 |
| LoRA | 23.39 | 15.20 | 13.28 |
| SumRA (freeze matrix $A$) | 20.77 | 14.49 | 13.39 |
| SumRA (train matrix $A$) | **20.14** | **13.75** | **13.02** |

adding a small set of domain-specific terms—this broad update mechanism offers less benefit, as changes are required for only a limited subset of the vocabulary.

Consistent with this intuition, our preliminary experiments show that SumRA does not yield substantial improvements when adapting LLaMA models Touvron et al. (2023) to GLUE tasks Wang et al. (2018). We hypothesize that these tasks primarily require learning task-specific decision boundaries over existing representations, rather than adapting global model attributes. As they do not involve broad shifts in representation space (e.g., accent or speaking style), they resemble localized adaptations, where SumRA's coordinated global update mechanism provides limited advantage.

## 7 CONCLUSION

We introduced a principled initialization strategy for LoRA that leverages singular vectors from pretraining to improve adaptation. By summing orthogonal components and freezing A, our method reduces trainable parameters while enhancing performance on low-resource multilingual ASR tasks.

ACKNOWLEDGMENTS

This research is supported by the National Research Foundation, Singapore and Infocomm Media Development Authority under its Trust Tech Funding Initiative. Any opinions, findings and conclusions or recommendations expressed in this material are those of the author(s) and do not reflect the views of National Research Foundation, Singapore and Infocomm Media Development Authority.

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

## A APPENDIX

### A.1 PROOF FOR THE OPTIMALITY OF THE GREEDY ASSIGNMENT

**Theorem 1.** Let there be $K$ right singular vectors in $V^T$, each associated with a singular value $I_1, I_2, \ldots, I_K$, sorted in non-increasing order:

$$I_1 \geq I_2 \geq \cdots \geq I_K > 0$$

Let there be $r$ rows in the LoRA matrix $A$, where $r \leq K$. Each singular vector must be assigned to exactly one row in $A$. For any row $j \in \{1, \ldots, r\}$, define its load $L_j$ as the sum of the singular values of all singular vectors assigned to that row:

$$L_j = \sum_{i \in S_j} I_i$$

where $S_j$ is the set of singular vectors assigned to row $j$. Let the cost of an assignment be the maximum row load:

$$L_{\max} = \max_j L_j$$

Then, the greedy algorithm that, at each step, assigns the next singular vector (in decreasing order of singular value) to the row with the currently minimum load, produces an assignment that minimizes $L_{\max}$. That is, the greedy algorithm is optimal.

*Proof.* We first observe that any assignment must satisfy the following lower bound on the maximum row load. Given that $L_{\max}^*$ is the maximum row load for the optimal assignment:

$$L_{\max}^* \geq \max \left( \frac{1}{r} \sum_{i=1}^{K} I_i, \ I_1 \right)$$

This is because the total singular value $\sum_{i=1}^{K} I_i$ must be distributed across the $r$ rows, and the largest singular value $I_1$ must be assigned to some row, which then has a load of at least $I_1$.

Let $L_{\max}^G$ denote the maximum row load produced by the greedy algorithm. We aim to show that:

$$L_{\max}^G = L_{\max}^*$$

We prove this by induction on $K$, the number of singular vectors.

1. *Base Case* ($K = r$): Each singular vector is assigned to exactly one row of the LoRA matrix $A$. Since the singular values are sorted in descending order and the algorithm always assigns the next largest singular value to the currently lightest row (which are all initially empty), each row receives exactly one singular vector. The load of the heaviest row in this case is exactly $I_1$, and no other assignment can result in a smaller maximum load, since $I_1$ must appear in some row. Thus, the greedy assignment is optimal.

2. *Inductive Step* (Case $K = m + 1$): Assume that the greedy algorithm produces an optimal assignment for any instance with $K = m$ singular vectors and $r$ rows, where $m \geq r$. The greedy algorithm assigns the first $m$ singular vectors optimally by the inductive hypothesis. It then assigns $I_{m+1}$, the next largest singular value, to the row with the smallest current load.

   Suppose, for contradiction, that there exists another assignment that produces a smaller maximum row load than the greedy one. Denote this alternative assignment by $Q$, and its maximum row load by $L_{\max}^Q < L_{\max}^G$. Let $j^*$ be the row to which $I_{m+1}$ is assigned under the greedy algorithm, and let $L_{j^*}^G$ be its load after this assignment. By construction, $j^*$ has the minimum load among all rows prior to assigning $I_{m+1}$, and therefore assigning $I_{m+1}$ to any other row (as might happen in $Q$) would result in a post-assignment load that is at least as large, if not larger than $L_{j^*}^G$. In the case that $L_{j^*}^G$ becomes the maximum row load after greedy assignment, the alternative assignment $Q$ must therefore result in a maximum row load equal to or larger than $L_{j^*}^G$. So, the assumption that the assignment $Q$ can improve upon the greedy assignment to yield a strictly lower maximum row load leads to a contradiction. So by contradiction, the theorem holds for $K = m + 1$.

By induction, the greedy algorithm always produces an assignment that minimizes the maximum row load for any $K \geq r$. $\qquad \square$

