# OpenReview forum: "SumRA: Parameter Efficient Fine-tuning with Singular Value Decomposition and Summed Orthogonal Basis"
_ICLR.cc/2026/Conference — ICLR 2026 Poster_

### Official Review · Reviewer_VaDY · 2025-10-23

**Soundness:** 2
**Presentation:** 3
**Contribution:** 2
**Rating:** 4
**Confidence:** 4

**Summary:**

This paper proposes SumRA, a parameter-efficient fine-tuning (PEFT) method that improves upon PiSSA by initializing each row of the LoRA matrix A as a sum of multiple singular vectors from the SVD of pretrained weights, rather than using only the leading singular vectors. The key insight is that PiSSA's use of only top-r singular vectors restricts adaptation to a narrow knowledge subspace. By summing multiple singular vectors per row (with proposed interleave-sum and greedy-sum strategies), SumRA aims to influence a broader portion of the model's knowledge space. The method is evaluated on multilingual ASR tasks using Whisper, showing improvements over LoRA baselines while using 50% fewer trainable parameters by freezing matrix A.

**Strengths:**

1. **Clear motivation**: The limitation of PiSSA using only top singular vectors is well-articulated, and the proposed solution is intuitive.

2. **Comprehensive baselines**: The paper compares against multiple LoRA variants (LoRA, LoRA-FA, PiSSA, CorDA, DoRA, VeRA).

3. **Consistent improvements**: Results show consistent WER improvements across multiple languages and model sizes (up to 16% relative improvement).

4. **Mathematical rigor**: The proof of greedy sum optimality (Appendix A.1) is mathematically sound, even if its connection to performance is unclear.

5. **Reproducibility**: Training details and hyperparameters are clearly specified.

**Weaknesses:**

1. **Weak theoretical foundation**: The paper lacks evidence that singular vectors encode vocabulary-specific knowledge. This is the central premise but is not validated. The authors should:
   - Provide empirical analysis of what different singular vectors represent
   - Show correlation between singular vector indices and vocabulary subsets
   - Validate the "concept encoding" hypothesis with interpretability experiments

2. **Incomplete experimental analysis**:
   - No analysis of which singular vectors are being summed and whether they truly represent complementary knowledge
   - Missing ablation: What if A is trained instead of frozen?
   - No visualization of learned representations to validate the broader knowledge influence claim
   - Limited to speech; NLP experiments needed to claim generality

3. **Unfair comparisons**:
   - SumRA uses SVD initialization while baseline LoRA uses random initialization
   - Should compare against: (a) PiSSA with frozen A, (b) SumRA with trained A
   - Storage cost comparison (Figure 4) is misleading—the savings come from freezing A, not from summing

4. **Statistical significance**: No error bars, confidence intervals, or multiple runs reported. With improvements sometimes <1% WER, statistical testing is essential.

5. **Limited scope**:
   - Only evaluated on low-resource (10h) settings; unclear if benefits hold with more data
   - Only one model architecture family
   - No analysis of computational cost during training/inference

6. **Interference analysis missing**: The paper mentions "destructive interference" between summed vectors but provides no quantitative analysis of this phenomenon or how much information is actually lost.

**Questions:**

1. What happens if you train A instead of freezing it? Is the improvement from initialization or from freezing?

2. Can you provide empirical evidence that different singular vectors encode different vocabulary subsets in speech models?

3. How does performance scale with the amount of training data (10h vs. 100h vs. full data)?

4. What is the computational overhead of SVD during initialization, especially for large models?

5. How do you choose which singular vectors to sum? Is there an automatic way to determine optimal subsets?

6. Can you show t-SNE or other visualizations demonstrating that SumRA representations cover broader knowledge space than PiSSA?

7. Why does greedy sum sometimes perform worse than interleave sum (Table 3, languages mhr and kmr)?

8. How sensitive is the method to the rank r? What happens with very small (r=1) or very large (r=64) ranks?

---

> ### Author Response · Authors · 2025-11-20
>
> > Response to weakness: Weak theoretical foundation:
> Our work is related to theoretical foundations like NdLinear [1], which addresses preserving multi-dimensional structure in parameter-efficient methods, relevant to the claimed broader knowledge coverage as mentioned by reviewer ckmZ. We promise to discuss these in the paper to give more theoretical justification.
>
>
>
> > Response to weakness: Incomplete experimental analysis:
> 1.	No analysis of which singular vectors are being summed and whether they truly represent complementary knowledge: The vectors may not be complementary. Some vectors may be important, and some vectors may be irrelevant. By summing all the vectors, our method avoids losing information (vector) important to the adapted task.
> 2.	Missing ablation: What if A is trained instead of frozen?:
> The table below shows the results of adapting Whisper to a new language Esperanto using three data scales (10h, 50h and 100h). The results show that training A also (so A is task-specific) gives slightly better performance in SumRA, but at the cost of more trainable parameters.
> | Method | 10h   | 50h   | 100h  |
> |--------|-------|-------|-------|
> | FT   | 18.89 | 15.31 | 13.62 |
> | LoRA   | 23.39 | 15.20 | 13.28 |
> | SumRA  | 20.77 | 14.49 | 13.39 |
> | SumRA train A  | 20.14 | 13.75 | 13.02 |
> 3.	No visualization of learned representations to validate the broader knowledge influence claim: Our work is related to theoretical foundations like NdLinear [1], relevant to the claimed broader knowledge coverage as mentioned by reviewer ckmZ.
> 4.	Limited to speech: We acknowledge that our evaluation focuses on the audio domain, but we believe LoRA’s applicability for user personalisation is more critical in the audio domain, as audio systems need to be fine-tuned to handle variability in language, accent, speaking style, and recording conditions (and their permutation) [2-7]. In contrast, LLM user personalization can often be addressed with LLM prompting, and we believe vision model user personalization is less relevant.
>
>
>
> > Response to weakness: Unfair comparisons:
> 1.	SumRA uses SVD initialization while baseline LoRA uses random initialization: Our method tried to show that SVD initialization (with summation) is better than random initialization.
> 2.	Should compare against: (a) PiSSA with frozen A, (b) SumRA with trained A:
> >>a)	PiSSA with frozen A is worse than PiSSA
> | Method |eo   | ia   | fy  |
> |--------|-------|-------|-------|
> | PiSSA  | 22.87 | 15.41 | 39.62 |
> | PiSSA with frozen A  | 25.71 | 16.11 | 43.23 |
> >b)	SumRA with trained A: please refer to the "Incomplete experimental analysis" section.
> 3.	Figure 4 is misleading: Thank you for pointing out! We will remove the summation sign for clarity.
> 4.	Statistical significance: We agree that error bars with further improve statistical significance. However, despite not having error bars, we have shown WERs of over 200 experiments in Table 2, showing consistent results.
>
>
>
> [1] Alex Reneau, Jerry Yao-Chieh Hu, Zhongfang Zhuang, Ting-Chun Liu, Xiang He, Judah Goldfeder, Nadav Timor, Allen G Roush, Ravid Shwartz-Ziv: NdLinear: Preserving Multi-Dimensional Structure for Parameter-Efficient Neural Networks. arXiv:2503.17353.
>
> [2] BN, Suhas, et al. "Fine-Tuning Large Audio-Language Models with LoRA for Precise Temporal Localization of Prolonged Exposure Therapy Elements." arXiv preprint arXiv:2506.09707 (2025).
>
> [3] Bagat, Raphaël, Irina Illina, and Emmanuel Vincent. "Mixture of LoRA Experts for Low-Resourced Multi-Accent Automatic Speech Recognition." arXiv preprint arXiv:2505.20006 (2025).
>
> [4] Zheng, Xinhu, et al. "Improving anomalous sound detection via low-rank adaptation fine-tuning of pre-trained audio models." 2024 IEEE Spoken Language Technology Workshop (SLT). IEEE, 2024.
>
> [5] Kwon, Ki-Joong, Jun-Ho So, and Sang-Hoon Lee. "Parameter-Efficient Fine-Tuning for Low-Resource Text-to-Speech via Cross-
> Lingual Continual Learning." Proc. Interspeech. Vol. 2025. 2025.
>
> [6] Ma, Yujian, Jinqiu Sang, and Ruizhe Li. "Behind the Scenes: Mechanistic Interpretability of LoRA-adapted Whisper for Speech Emotion Recognition." arXiv preprint arXiv:2509.08454 (2025).
>
> [7] Joseph, George, and Arun Baby. "Speaker personalization for automatic speech recognition using weight-decomposed low-rank adaptation." Proc. Interspeech. 2024.
>
> [8] Peng, Jing, et al. "A survey on speech large language models for understanding." Authorea Preprints (2025).
>
> [9] Zhang, Longteng, et al. "Lora-fa: Memory-efficient low-rank adaptation for large language models fine-tuning." arXiv preprint arXiv:2308.03303 (2023).

---

> ### Author Response · Authors · 2025-11-20
>
> > Response to weakness: Limited scope
> 1.	Only evaluated on low-resource (10h) settings: Please allow us to explain that in our user personalization scenario, the training data will mostly be low resource. However, to broaden the experimental setting, we additionally evaluate SumRA under different data scales (10h, 50h, and 100h) as shown in the table above.
> 2.	Only one model architecture family: Please allow us to explain that the transformer-based encoder-decoder architecture we tested represents a main category of ASR models, and it is the main architecture for large speech foundation models [8].
> 3.	No analysis of computational cost during training/inference: The inference cost is theoretically the same as LoRA as the model architecture is the same. The computation cost during training has been extensively studied by a similar work LoRA-FA [9].
> 4.	Interference analysis missing: Empirically, our ablation study in Table 3 of the paper shows consistent results to prove that block sum is worse than greedy sum. This suggests that destructive interference of important singular vectors plays a key role in SumRA fine-tuning.
>
> >Q1: What happens if you train A instead of freezing it? Is the improvement from initialization or from freezing?:
> Our previous discussion (Incomplete experimental analysis section point 2) show that the improvement comes from initialization instead of freezing.
>
> >Q2: Can you provide empirical evidence that different singular vectors encode different vocabulary subsets in speech models?
> Although we have not provided empirical evidence, another theoretical foundation work NdLinear [1] addresses preserving multi-dimensional structure in parameter-efficient methods, which provides theoretical justification to the claimed broader knowledge coverage.
>
> >Q3: How does performance scale with the amount of training data (10h vs. 100h vs. full data)?
> To broaden the experimental setting, we additionally evaluate SumRA under different data scales (10h, 50h, and 100h) when adapting Whisper to the new language Esperanto (this new language has more training data) as shown in the table above.
>
> >Q4: What is the computational overhead of SVD during initialization, especially for large models?:
> The computational overhead is negligible compared to the training cost as computing SVD is quite fast.
>
> >Q5: How do you choose which singular vectors to sum? Is there an automatic way to determine optimal subsets?:
> Our method proposes two strategies, which are interleave sum and greedy sum to automatically select singular vectors.
>
> >Q6: Can you show t-SNE or other visualizations demonstrating that SumRA representations cover broader knowledge space than PiSSA?:
> Although we cannot show t-SNE at the moment, the idea that using more singular vectors can cover broader knowledge space is theoretically backed by NDLinear [1].
>
>
> >Q7: Why does greedy sum sometimes perform worse than interleave sum (Table 3, languages mhr and kmr)?:
> We believe interleave sum and greedy sum can balance the importance values well enough as shown in Figure 3 of the paper, so they share similar performance.
>
> >Q8: How sensitive is the method to the rank r? What happens with very small (r=1) or very large (r=64) ranks?:
> We have tested the sensitivity of r=2 and r=32 in Table 2 of the paper. We believe the results of r=1 should be close to r=2, and the results of r=64 should be close to r=32.
>
> [1] Alex Reneau, Jerry Yao-Chieh Hu, Zhongfang Zhuang, Ting-Chun Liu, Xiang He, Judah Goldfeder, Nadav Timor, Allen G Roush, Ravid Shwartz-Ziv: NdLinear: Preserving Multi-Dimensional Structure for Parameter-Efficient Neural Networks. arXiv:2503.17353.
>
> [8] Peng, Jing, et al. "A survey on speech large language models for understanding." Authorea Preprints (2025).
>
> [9] Zhang, Longteng, et al. "Lora-fa: Memory-efficient low-rank adaptation for large language models fine-tuning." arXiv preprint arXiv:2308.03303 (2023).

---

### Official Review · Reviewer_Q2k7 · 2025-10-31

**Soundness:** 3
**Presentation:** 3
**Contribution:** 2
**Rating:** 6
**Confidence:** 4

**Summary:**

This paper can be seen as an upgrade of PiSSA and LoRA.
To address the issue in PiSSA that “A is typically initialized with only the leading singular vectors, which limits its representational capacity to a narrow subspace of the model’s knowledge,”
the paper proposes SumRA, which initializes each row of A as a sum of multiple singular vectors selected beyond the leading components, while B is initialized to zero, as in LoRA.
Experimental results on multilingual automatic speech recognition tasks show that SumRA outperforms many state-of-the-art fine-tuning methods.

**Strengths:**

1. The idea of initializing each row of A as a sum of multiple singular vectors chosen from beyond the leading components—thereby enabling A to influence a larger portion of the model’s knowledge space—is quite intuitive.

2. The writing of the paper is very fluent and easy to read.

3. In terms of experimental results, SumRA achieves significant improvements compared to various other methods.

**Weaknesses:**

1. This paper conducts experiments only on ASR tasks using the Whisper series of models.
However, the compared methods such as LoRA, DoRA, PiSSA, and CorDA mostly focus on fine-tuning large language models (LLMs).
Therefore, it is necessary to validate the effectiveness of SumRA on LLMs to enhance the credibility of the experimental results.

2. Although the motivation behind SumRA is quite intuitive, the paper does not sufficiently reveal the underlying mechanism that explains why it works.
It would be helpful to include some theoretical justification or experimental analysis to support the claim that
initializing each row of A as a sum of multiple singular vectors chosen from beyond the leading components
indeed enables A to influence a larger portion of the model’s knowledge space.

3. In Figure 1, the depiction of PiSSA’s initialization of W_0 is incorrect; this initialization would break the model’s capability at the start.

**Questions:**

See Weaknesses

---

> ### Author Response · Authors · 2025-11-20
>
> > Response to weakness: This paper conducts experiments only on ASR tasks LoRA, DoRA, PiSSA, and CorDA, designed for LLM tasks:
> We agree that LoRA methods are originally published as methods to fine-tune large language models (LLMs). However, we believe LoRA’s applicability for user personalisation is more critical in the audio domain, as audio systems need to be fine-tuned to handle variability in language, accent, speaking style, and recording conditions (and their permutation) [1-6]. In contrast, LLM user personalization can often be addressed with LLM prompting.
>
>
>
> > Response to weakness: It would be helpful to include some theoretical justification on why the model works:
> We agree that we need to discuss how our work brings connection with several pertinent works which includes Hu et al. [7], SORSA [8], NdLinear [9], and recent theoretical foundations on fine-tuning limits [10,11] as mentioned by reviewer ckmZ. We promise to discuss these in the paper to give more theoretical justification.
>
>
>
>
> > Respone to weakness: In Figure 1, the depiction of PiSSA’s initialization of W_0 is incorrect; this initialization would break the model’s capability at the start:
> Yes, we agree. Thank you for pointing out the typo in the Figure!
>
>
>
> [1] BN, Suhas, et al. "Fine-Tuning Large Audio-Language Models with LoRA for Precise Temporal Localization of Prolonged Exposure Therapy Elements." arXiv preprint arXiv:2506.09707 (2025).
>
> [2] Bagat, Raphaël, Irina Illina, and Emmanuel Vincent. "Mixture of LoRA Experts for Low-Resourced Multi-Accent Automatic Speech Recognition." arXiv preprint arXiv:2505.20006 (2025).
>
> [3] Zheng, Xinhu, et al. "Improving anomalous sound detection via low-rank adaptation fine-tuning of pre-trained audio models." 2024 IEEE Spoken Language Technology Workshop (SLT). IEEE, 2024.
>
> [4] Kwon, Ki-Joong, Jun-Ho So, and Sang-Hoon Lee. "Parameter-Efficient Fine-Tuning for Low-Resource Text-to-Speech via Cross-Lingual Continual Learning." Proc. Interspeech. Vol. 2025. 2025.
>
> [5] Ma, Yujian, Jinqiu Sang, and Ruizhe Li. "Behind the Scenes: Mechanistic Interpretability of LoRA-adapted Whisper for Speech Emotion Recognition." arXiv preprint arXiv:2509.08454 (2025).
>
> [6] Joseph, George, and Arun Baby. "Speaker personalization for automatic speech recognition using weight-decomposed low-rank adaptation." Proc. Interspeech. 2024.
>
> [7] Jerry Yao-Chieh Hu, Maojiang Su, En-Jui Kuo, Zhao Song, Han Liu: Computational Limits of Low-Rank Adaptation (LoRA) Fine-Tuning for Transformer Models. ICLR 2025.
>
> [8] Yang Cao, Zhao Song: SORSA: Singular Values and Orthonormal Regularized Singular Vectors Adaptation of Large Language Models. arXiv:2409.00055.
>
> [9] Alex Reneau, Jerry Yao-Chieh Hu, Zhongfang Zhuang, Ting-Chun Liu, Xiang He, Judah Goldfeder, Nadav Timor, Allen G Roush, Ravid Shwartz-Ziv: NdLinear: Preserving Multi-Dimensional Structure for Parameter-Efficient Neural Networks. arXiv:2503.17353.
>
> [10] Timothy Chu, Zhao Song, Chiwun Yang: Fine-tune Language Models to Approximate Unbiased In-context Learning. arXiv:2310.03331.
>
> [11] Jerry Yao-Chieh Hu, Wei-Po Wang, Ammar Gilani, Chenyang Li, Zhao Song, Han Liu: Fundamental Limits of Prompt Tuning Transformers: Universality, Capacity and Efficiency. arXiv:2411.16525.

---

> > ### Comment · Reviewer_Q2k7 · 2025-11-26
> >
> > Thank you for the authors’ response. My concerns have been largely addressed, so I will keep my score as Rating: 6, a positive evaluation. However, I still recommend that the authors include LLM experiments in the final version. I acknowledge that the ASR experiments are valuable, and I understand the authors’ point that LLM user personalization can often be achieved through prompting. Nevertheless, the baseline methods compared in the paper are all designed for LLMs. Adding such experiments would strengthen the credibility of the work.

---

> > > ### Author Response · Authors · 2025-11-26
> > >
> > > Dear Reviewer,
> > >
> > > Thank you so much for your positive comments in our work. As suggested,  we will include LLM experiments in our final version. Thanks again.
> > >
> > > Best regards,
> > >
> > > All authors

---

### Official Review · Reviewer_ckmZ · 2025-10-31

**Soundness:** 3
**Presentation:** 2
**Contribution:** 3
**Rating:** 6
**Confidence:** 4

**Summary:**

This paper proposes SumRA, a parameter-efficient fine-tuning method that improves upon PiSSA by initializing each row of the LoRA A matrix as a sum of multiple singular vectors from the SVD decomposition of pretrained weights, rather than using only the top-r singular vectors. The authors freeze matrix A during training and only update matrix B, reducing trainable parameters by 50% compared to standard LoRA. They introduce three summation strategies (block sum, interleave sum, and greedy sum) to distribute singular vectors across rows while minimizing interference between important components. Experiments on multilingual ASR adaptation using Whisper show that SumRA achieves 12-16% relative WER improvement over LoRA baselines across five low-resource languages with only 10 hours of training data each, while using half the trainable parameters.

**Strengths:**

1. Well-motivated approach with clear theoretical foundation. The paper provides strong intuition for why summing multiple singular vectors enables broader knowledge coverage compared to PiSSA's approach of using only top-r vectors (Figure 2D and Section 3.1). The connection to model averaging in Section 4 further strengthens the conceptual framework, and the formal proof of greedy sum optimality in Appendix A.1 adds mathematical rigor to the method.
2. Comprehensive experimental validation with consistent improvements. The experiments systematically compare SumRA against multiple baselines (LoRA, LoRA-FA, VeRA, DoRA, PiSSA, CorDA) across two model sizes (Whisper-small and Whisper-large-v2), two ranks (2 and 32), and five languages. SumRA consistently outperforms all baselines, achieving improvements across nearly all experimental configurations while using 50% fewer trainable parameters, demonstrating both effectiveness and practical efficiency.

**Weaknesses:**

1. Limited evaluation domains. The paper exclusively evaluates on multilingual ASR adaptation with only 10 hours of training data per language from Common Voice. This narrow experimental setting raises concerns about generalizability. No experiments are provided to validate this claim or explore other domains such as natural language understanding, vision tasks, or even ASR with different data scales (e.g., 100 hours, 1000 hours). The evaluation should be expanded to include at least one additional modality (e.g., text-based tasks using LLaMA or similar models) and different data regimes to demonstrate broader applicability.

2. Incomplete comparison with related work. The paper lacks comparisons with several directly relevant PEFT methods and omits discussion of recent theoretical advances that would contextualize the contributions. Experimentally, the paper does not compare against AdaLoRA [1], which adaptively allocates ranks and could demonstrate whether fixed summation outperforms adaptive strategies, or LoRA+ [2], which improves LoRA through differential learning rates for A and B. Theoretically and methodologically, the paper misses several pertinent works: Hu et al. [3] establish computational limits of LoRA that could inform when summation strategies are beneficial; SORSA [4] uses orthonormal regularization of singular vectors, directly related to this work's orthogonality claims (lines 264-265); NdLinear [5] addresses preserving multi-dimensional structure in parameter-efficient methods, relevant to the claimed broader knowledge coverage; and recent theoretical foundations on fine-tuning limits [6,7] could strengthen the motivation for summing singular vectors. These omissions make it difficult to assess whether SumRA advances beyond current state-of-the-art or whether existing methods achieve similar benefits.

3. Insufficient analysis of the summation mechanism. While the paper provides intuition about reducing "destructive interference" between singular vectors (lines 270-271, 283), there is no empirical analysis of what actually happens when vectors are summed.

# Reference

[1]: Qingru Zhang, Minshuo Chen, Alexander Bukharin, Nikos Karampatziakis, Pengcheng He, Yu Cheng, Weizhu Chen, Tuo Zhao: AdaLoRA: Adaptive Budget Allocation for Parameter-Efficient Fine-Tuning. ICLR 2023.

[2] Hayou, Soufiane, Nikhil Ghosh, Bin Yu: Lora+: Efficient low rank adaptation of large models. arXiv preprint arXiv:2402.12354.

[3] Jerry Yao-Chieh Hu, Maojiang Su, En-Jui Kuo, Zhao Song, Han Liu:
Computational Limits of Low-Rank Adaptation (LoRA) Fine-Tuning for Transformer Models. ICLR 2025.

[4] Yang Cao, Zhao Song: SORSA: Singular Values and Orthonormal Regularized Singular Vectors Adaptation of Large Language Models. arXiv:2409.00055.

[5] Alex Reneau, Jerry Yao-Chieh Hu, Zhongfang Zhuang, Ting-Chun Liu, Xiang He, Judah Goldfeder, Nadav Timor, Allen G Roush, Ravid Shwartz-Ziv: NdLinear: Preserving Multi-Dimensional Structure for Parameter-Efficient Neural Networks. arXiv:2503.17353.

[6] Timothy Chu, Zhao Song, Chiwun Yang: Fine-tune Language Models to Approximate Unbiased In-context Learning. arXiv:2310.03331.

[7] Jerry Yao-Chieh Hu, Wei-Po Wang, Ammar Gilani, Chenyang Li, Zhao Song, Han Liu: Fundamental Limits of Prompt Tuning Transformers: Universality, Capacity and Efficiency. 	arXiv:2411.16525.

**Questions:**

1. Have you conducted any preliminary experiments on text-based fine-tuning like instruction-tuning LLMs or vision tasks that you could share?

2. Can you provide experimental evidence for this multi-task scenario? What is the actual performance when you train one shared A matrix with task-specific B matrices across all 5 languages compared to task-specific (A,B) pairs?

3. Have you measured the actual information loss when summing singular vectors, for example by comparing reconstruction error?

---

> ### Author Response · Authors · 2025-11-20
>
> > Response to weakness: Limited evaluation domains:
> To broaden the experimental setting, we additionally evaluate SumRA under different data scales (10h, 50h, and 100h) for language Esperanto (this new language has more training data). The results show that SumRA performs better in low-resource scenarios:
> | Method | 10h   | 50h   | 100h  |
> |--------|-------|-------|-------|
> | FT   | 18.89 | 15.31 | 13.62 |
> | LoRA   | 23.39 | 15.20 | 13.28 |
> | SumRA  | 20.77 | 14.49 | 13.39 |
> | SumRA train A  | 20.14 | 13.75 | 13.02 |
> We acknowledge that our evaluation focuses on the audio domain, but we believe LoRA’s applicability for user personalisation is more critical in the domain, as ASR systems needs to be fine-tuned to handle variability in language, accent, speaking style, and recording conditions (and their permutation) [1-5]. In contrast, LLM user personalization can often be addressed with LLM prompting, and we believe vision model user personalization is less relevant.
>
>
>
> > Incomplete comparison with related work:
> Comparison with AdaLoRA: Their work requires to iteratively prune singular values in correspondence to their importance score during the training. Our SumRA method is much simpler as we prune the weight matrix only once at the beginning of the training.
> Comparison with LoRA+: Our work is orthogonal to theirs as they focus on learning rate adjustments for LoRA based approaches. Our work focuses on better LoRA initialization methods.
> We agree that our work brings connection with several pertinent works which includes Hu et al. , SORSA, NdLinear, and recent theoretical foundations on fine-tuning limits as suggested. We promise to discuss these in the paper.
>
>
>
> > Insufficient analysis of the summation mechanism:
> Empirically, our ablation study in Table 3 of the paper shows consistent results to prove that block sum is worse than greedy sum. This suggests that destructive interference of important singular vectors plays a key role in SumRA fine-tuning.
>
>
>
> >Response to Q1: Have you conducted any preliminary experiments on text-based fine-tuning like instruction-tuning LLMs or vision tasks that you could share? :
> We currently focus only on the audio domain, which we believe is the most relevant scenario for our method. Furthermore, most existing research [1-5] has demonstrated that fine-tuning audio models with audio data is the most effective approach. Consequently, audio models are typically fine-tuned solely with audio data as model input and do not involve text or image fine-tuning data.
>
>
>
> > Response to Q2: Can you provide experimental evidence for this multi-task scenario? What is the actual performance when you train one shared A matrix with task-specific B matrices?:
> According to the table above, we empirically found that training A also (so A is task-specific) gives slightly better performance in SumRA, but at the cost of more trainable parameters.
>
>
>
> >Response to Q3: Have you measured the actual information loss when summing singular vectors, for example by comparing reconstruction error?:
> If we are not summing the singular vectors to reduce the matrix rank, we are essentially fine-tuning a full rank (not low rank) LoRA matrix. In this case the performance will be close to full fine-tuning (FT), where we have provided the results in Table 2 of the paper.
>
>
>
> [1] BN, Suhas, et al. "Fine-Tuning Large Audio-Language Models with LoRA for Precise Temporal Localization of Prolonged Exposure Therapy Elements." arXiv preprint arXiv:2506.09707 (2025).
>
> [2] Bagat, Raphaël, Irina Illina, and Emmanuel Vincent. "Mixture of LoRA Experts for Low-Resourced Multi-Accent Automatic Speech Recognition." arXiv preprint arXiv:2505.20006 (2025).
>
> [3] Zheng, Xinhu, et al. "Improving anomalous sound detection via low-rank adaptation fine-tuning of pre-trained audio models." 2024 IEEE Spoken Language Technology Workshop (SLT). IEEE, 2024.
>
> [4] Kwon, Ki-Joong, Jun-Ho So, and Sang-Hoon Lee. "Parameter-Efficient Fine-Tuning for Low-Resource Text-to-Speech via Cross-Lingual Continual Learning." Proc. Interspeech. Vol. 2025. 2025.
>
> [5] Ma, Yujian, Jinqiu Sang, and Ruizhe Li. "Behind the Scenes: Mechanistic Interpretability of LoRA-adapted Whisper for Speech Emotion Recognition." arXiv preprint arXiv:2509.08454 (2025).
>
> [6] Joseph, George, and Arun Baby. "Speaker personalization for automatic speech recognition using weight-decomposed low-rank adaptation." Proc. Interspeech. 2024.

---

### Meta-Review · Area_Chair_KPPw · 2025-12-22

**Summary:**

This paper presents SumRA, a novel PEFT method that initializes low-rank adapters using sums of singular vectors to capture broader knowledge from pretrained weights. Reviewers find the core idea intuitive and the results on multilingual ASR convincing, demonstrating clear improvements over strong baselines. However, acceptance is contingent on addressing major concerns: the scope must be expanded beyond ASR with validation on LLMs, the central hypothesis about knowledge coverage requires supporting empirical analysis, and comparisons must be strengthened with fairer ablations and relevant recent methods. With these revisions, the paper would make a solid contribution to the field.

**Reviewer Concerns:**

Though some concerns from reviewer VaDY may still be outstanding, the other concerns from the rest of reviewers have been addressed.

**Reviewer Scores:**

The reviewers ckMZ and Q2K7 will maintain the positive scores. The reviewer VaDY will likely raise the score.

---

### Decision · Program_Chairs · 2026-01-26

Accept (Poster)